# The impact of the COVID-19 pandemic on waiting times for elective surgery patients: A multicenter study

Mikko Uimonen[1]*, Ilari Kuitunen[2,3], Juha Paloneva[1,2], Antti P. Launonen[4], Ville Ponkilainen[1], Ville M. Mattila[4,5]

1 Department of Surgery, Central Finland Hospital Nova, Jyväskylä, Finland, 2 School of Medicine, University of Eastern Finland, Kuopio, Finland, 3 Mikkeli Central Hospital, Mikkeli, Finland, 4 Department of Orthopaedics, Tampere University Hospital, Tampere, Finland, 5 Faculty of Medicine and Health Technology, Tampere University, Tampere, Finland

* mikko.uimonen@ksshp.fi

## Abstract

**Data Availability Statement:** All relevant data are within the paper and its Supporting Information files.

### Background

A concern has been that health care reorganizations during the first COVID-19 wave have led to delays in elective surgeries, resulting in increased complications and even mortality. This multicenter study examined the changes in waiting times of elective surgeries during the COVID-19 pandemic in Finland.

### Methods

Data on elective surgery were gathered from three Finnish public hospitals for years 2017–2020. Surgery incidence and waiting times were examined and the year 2020 was compared to the reference years 2017–2019. The mean annual, monthly, and weekly waiting times were calculated with 95% confidence intervals (CI). The most common diagnosis groups were examined separately.

### Findings

A total of 88 693 surgeries were included during the study period. The mean waiting time in 2020 was 92.6 (CI 91.5–93.8) days, whereas the mean waiting time in the reference years was 85.8 (CI 85.1–86.5) days, resulting in an average 8% increase in waiting times in 2020. Elective procedure incidence decreased rapidly in the onset of the first COVID-19 wave in March 2020 but recovered in May and June, after which the surgery incidence was 22% higher than in the reference years and remained at this level until the end of the year. In May 2020 and thereafter until November, waiting times were longer with monthly increases varying between 7% and 34%. In gastrointestinal and genitourinary diseases and neoplasms, waiting times were longer in 2020. In cardiovascular and musculoskeletal diseases, waiting times were shorter in 2020.

**Funding:** The authors received no specific funding for this work.

**Competing interests:** The authors have declared that no competing interests exist.

## Conclusion

The health care reorganizations due to the pandemic have increased elective surgery waiting times by as much as one-third, even though the elective surgery rate increased by one-fifth after the lockdown.

## Introduction

The spread of COVID-19 led to a nationwide lockdown in Finland in March 2020. During the first pandemic wave, emergency department visits decreased, but emergency surgeries remained unchanged [1]. In preparation for the predicted surge in COVID-19 cases, public health care was reorganized and elective surgeries postponed. By summer, the first wave had abated. During the second wave in the fall, regional stepwise restrictions were used instead of lockdown. During this period, elective surgeries were not postponed and efforts were made to address the accumulated backlog in elective surgery. A concern has been that actions taken during the first wave have led to delays in elective surgeries, resulting in increased complications and even mortality [2–4].

This multicenter study examined the changes in waiting times of elective surgeries during the COVID-19 pandemic in Finland.

## Methods

Anonymized data on elective surgeries from 2017–2020 were collected from the electronic patient registers of three large Finnish public hospitals–Central Finland Hospital, Mikkeli Central Hospital, and Tampere University Hospital–covering approximately 900 000 citizens (1/6th of Finnish population). In Finland, health care is publicly funded and accessible for all citizens, and preoperative assessment is made in hospital outpatient clinics. According to Finnish law, elective surgeries must be performed within six months from the initial decision. Waiting times in days were calculated from the time interval between the date of the decision to operate and the date of the operation. To focus solely on elective surgery, operations within 14 days from the decision were excluded.

The weekly incidences and 95% CIs of elective surgeries were calculated for 2020 and the reference years using Poisson exact method. The total population within study hospitals' catchment area was obtained from Statistics Finland [5]. The mean annual, monthly, and weekly waiting times for elective surgeries with 95% confidence intervals (CI) were calculated for year 2020 and for commonly for the reference years 2017–2019. The weekly elective surgery incidence and waiting times in year 2020 were compared to the reference years. We focused on changes in waiting times during the nationwide lockdown period (March 16 to June 1) and during the period of regional restrictions (September onwards). In addition, waiting times were stratified by diagnosis groups. These groups included cardiovascular (ICD-10 I*), musculoskeletal (M*), gastrointestinal (K2-K9*) and genitourinary (N*) diseases, and neoplasms (C* and D0-4*). To illustrate the differences, ratios between weekly mean waiting time in 2020 and in the reference years were calculated by dividing waiting times in 2020 by those in the reference years. Statistical analysis was performed using R (4.0.3) statistical software. According to the Finnish research legislation and The Finnish National Board on Research Integrity, appointed by the Ministry of Education and Culture, ethical committee approval was not required due to register-based study design [6]. Due to the retrospective register-based study

design with completely anonymized and non-identifiable patient data, consents to participate in the study were not required as stated by Finnish law [7].

## Results

A total of 88 693 surgeries were included. After the beginning of the lockdown, elective procedures decreased rapidly but recovered in May and June (Fig 1). When the lockdown was lifted at the beginning of the summer vacation period, the elective surgery incidence was 22% higher than in the reference years and remained at this level until the end of the year.

After the start of the lockdown in March and April 2020, elective surgery waiting times were 10% and 16% shorter than in the reference years, respectively (Fig 2). However, in May 2020 and thereafter, waiting times were longer until November, with monthly increases varying between 7% and 34%. The mean waiting time in 2020 was 92.6 (CI 91.5–93.8) days, whereas the mean waiting time in the reference years was 85.8 (CI 85.1–86.5) days, resulting in an average 8% increase in waiting times in 2020.

The annual mean surgery waiting time in musculoskeletal diseases was 93 (CI 92–95) days in 2020 and 100 (CI 98–102) days in the reference years (Fig 3). In gastrointestinal and genitourinary diseases, waiting times increased after the lockdown (Fig 3). The annual mean increases in waiting times were 8% in gastrointestinal diseases (92 [CI 88–96] days in 2020 vs. 85 [CI 83–87] days in the reference years) and 19% in genitourinary diseases (112 [CI 108–115] days in 2020 vs. 94 [CI 92–95] days in the reference years). In surgeries due to cardiovascular diseases, there were an occasional decrease in the waiting times during the lockdown period after which the waiting times followed the reference years' level (Fig 3). The mean annual waiting time for cardiovascular surgeries in 2020 (85 [CI 80–90] days) was 3% lower than in the reference years (88 [CI 86–91] days).

In neoplasms, waiting times shortened at the beginning of lockdown, but between June and September they were 18% to 28% longer than in the reference years (Fig 3). For the year 2020,

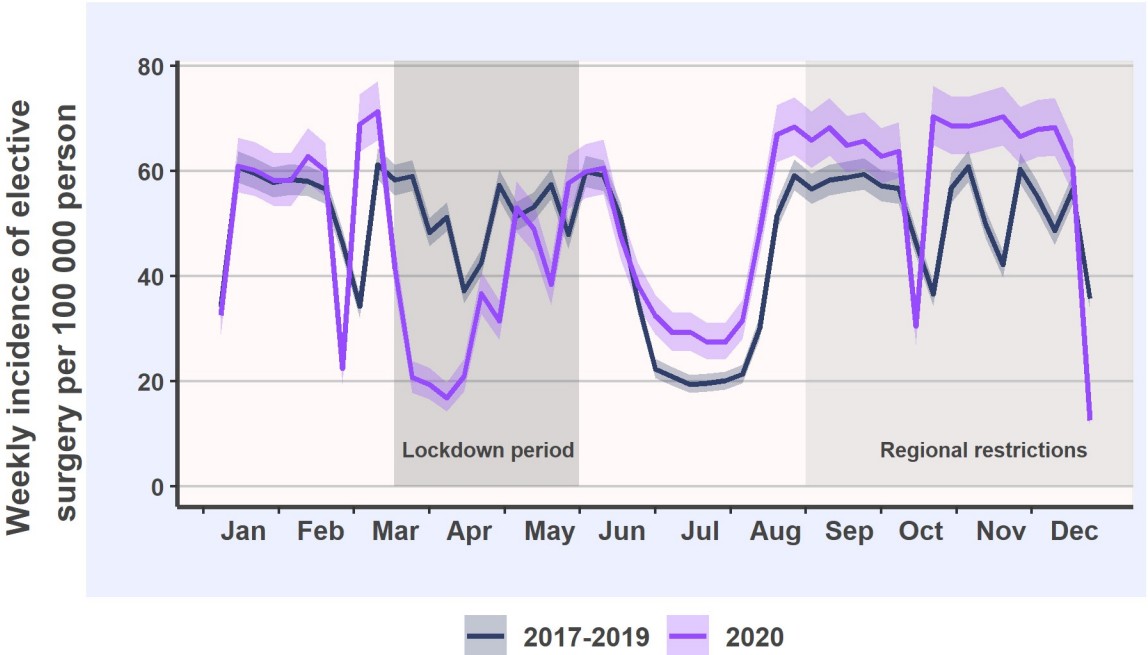

**Fig 1. Weekly incidence and 95% confidence intervals of elective surgeries in 2020 and in the reference years.**

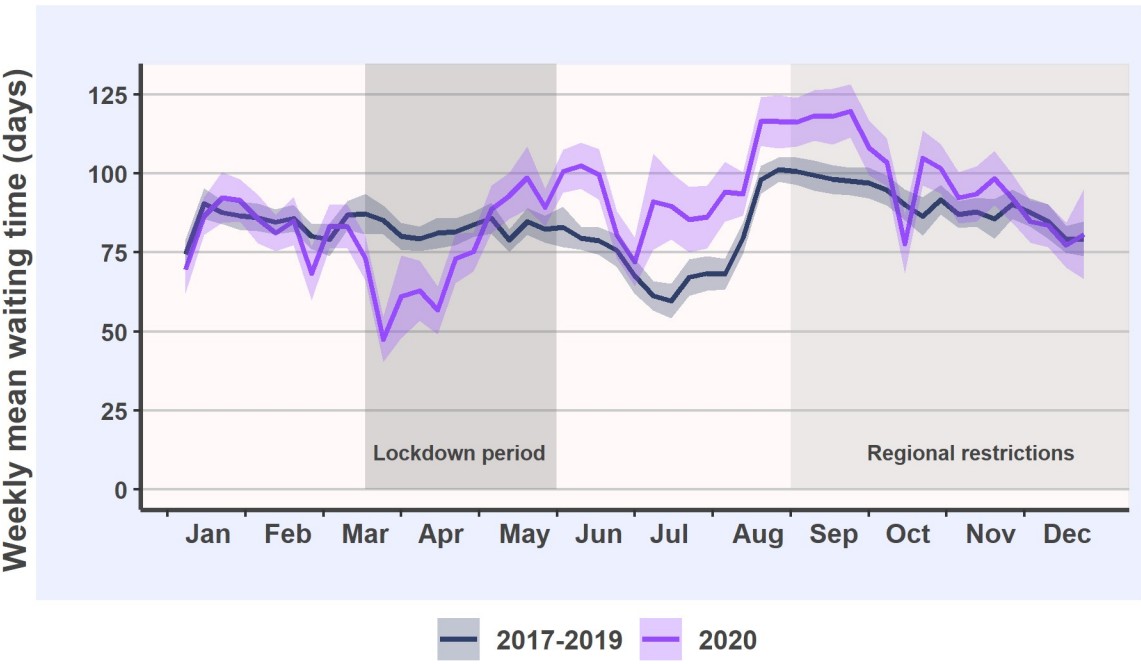

**Fig 2. Weekly mean and 95% confidence intervals of elective surgery waiting times in 2020 and the reference years.**

mean waiting times were 6% longer than in the reference years (59 [CI 57–61] days in 2020 vs. 56 [CI 54–57] days in the reference years).

## Discussion

According to our findings, COVID-19 had a substantial influence on waiting times for elective surgery. During lockdown, waiting times decreased temporarily, after which they increased rapidly and leveled-off until the end of 2020.

Concerns have been raised that COVID-19 would lead to delays in elective surgery due to the postponement of non-urgent procedures. Delays in surgery have been shown to have an impact on outcomes, with longer delays causing worse prognoses in many diseases. Thus, the benefit of surgery decreases along with longer waiting times [2, 8–10]. In the early phase of the pandemic, waiting times paradoxically decreased. This may be explained by the prioritization of surgeries. Nevertheless, the rapid increase in waiting times thereafter probably reflected an increasing treatment backlog in elective surgery. Although Finland did not experience severe intensive care unit overload or substantial pandemic, the lockdown and health care reorganizations led to a remarkable increase in waiting times.

Overall, the beginning of the lockdown period led to decreased elective surgery incidence and simultaneously decreased elective surgery waiting times. The magnitude of the decreases was similar to the decreases during summer vacation period observed during the reference years. In contrast to the reference years, in 2020 the elective surgery incidence during the summer vacation period and thereafter was higher until the end of the year. However, the waiting times increased above the reference level in May and remained high until October. These findings suggest that despite the considerable efforts made to solve the burden of cumulated elective surgery backlog during the summer period, it required at least five months to reach the reference levels of waiting times.

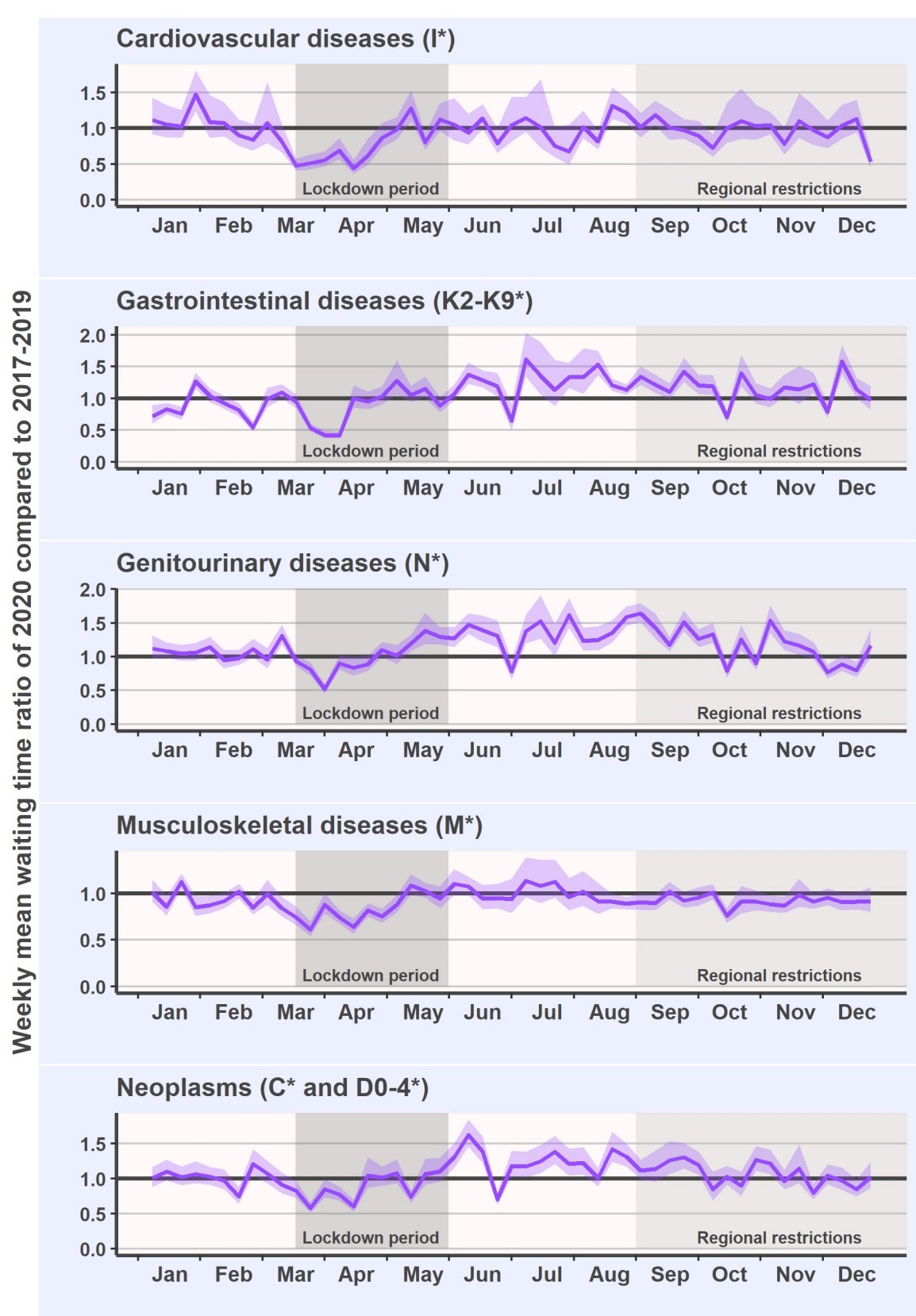

**Fig 3. Ratio between the weekly mean waiting time and 95% confidence intervals in 2020 (purple line) and the mean waiting time in the reference years (black parallel line) in elective surgeries of the most common diagnosis groups.**

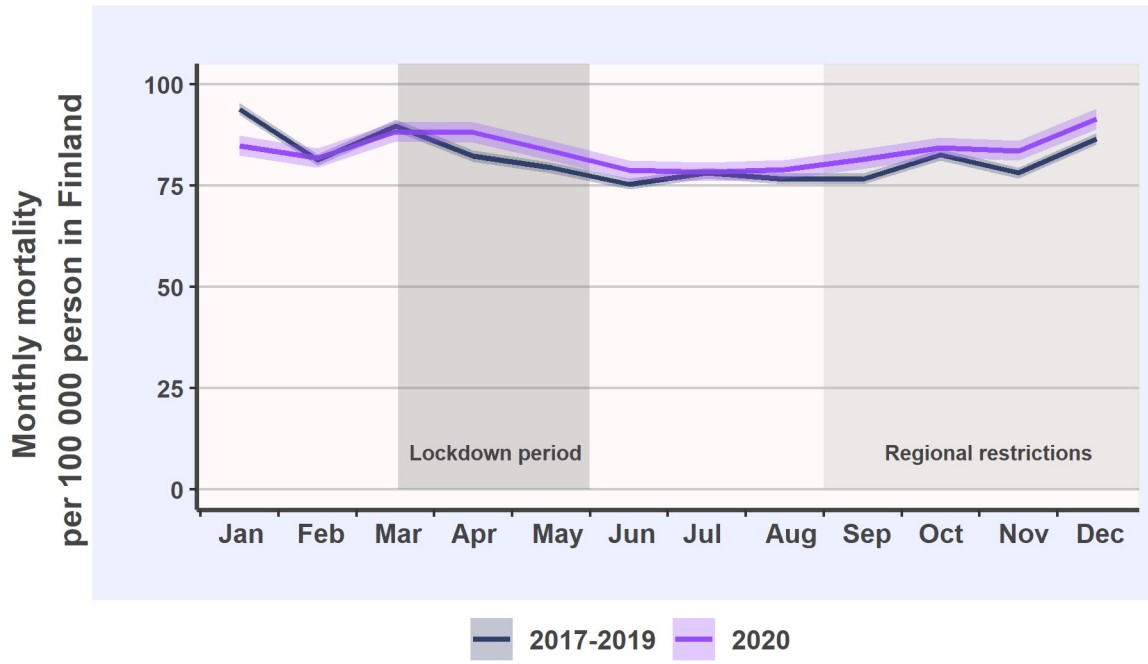

**Fig 4. Monthly national mortality incidence per 100 000 persons in Finland and 95% confidence intervals in 2020 (purple line) and in the reference years (black line).** Statistics Finland Population data, available in [https://pxnet2.stat.fi/PXWeb/pxweb/fi/StatFin/].

Examination of the diagnosis groups showed varying patterns between groups. In cardiovascular and musculoskeletal surgeries, the waiting times decreased occasionally during the lockdown and rapidly returned to the reference level. For the entire year, surgery waiting times were shorter in cardiovascular and musculoskeletal surgery in 2020 than in the reference years. In gastrointestinal and genitourinary diseases as well as neoplasms, waiting times were longer. In these diseases, the lengthened waiting times recovered after several months. In oncological surgery, delays may be detrimental for prognosis and survival [8, 10, 11]. However, a previous study from Finland reported that the rate of oncological surgeries remained stable during the pandemic despite the cancellations [12].

The consequences of COVID-19 pandemic and the limited elective capacity during the initial harsh restrictions may become more apparent in the near future in the form of longer treatment queues which may lead to worsening of diseases and increased mortality due to delayed diagnostics and treatment. In addition, due to the limited access to primary care during the initial lockdown, there may still be treatment backlog to be solved. Indeed, according to the national mortality data obtained from Statistics Finland database, overall all-cause mortality increased over the reference level after the initiation of the pandemic in Finland (Fig 4). The observed increase resulted in 4% higher mortality during March to December 2020 than during the corresponding period of the reference years (IRR 1.04 [CI 1.03–1.05]). Further, it is likely that the overall consequences of the pandemic on the mortality have not yet been seen but they may become visible over the following years.

To address and manage the treatment backlog in surgical units, traditional perioperative care protocols may need to be revised to enhance the patient flow and hence respond to the cumulated burden [13]. In future, carefully designed preparation plans for national emergencies with a possibility of rapid response to changing circumstances as well as proper restrictions enacted by the authorities and politicians are needed to secure the capacity and

procedure rates in surgery units, despite the pandemic. The COVID-19 pandemic should be considered as an important lesson for future policy making during possible future pandemics and similar conditions. The current pandemic plans were aimed to secure resources for managing the forthcoming COVID-19 surge but the plans how to mitigate and solve the cumulative treatment backlog and how to prioritize the resources were insufficient [14].

The main strength of this study is the representative Finnish public health care data collected from three large Finnish hospitals that included reference data from three previous years. The main limitation of this study is the common shortcomings of registries that in general eliminate the possibility to estimate the effects of waiting times on individual patients.

In summary, the health care reorganizations due to COVID-19 have increased elective surgery waiting times by as much as one-third, even though the elective surgery incidence increased by one-fifth after the lockdown. Delays were seen in procedures that were performed a few months after the beginning of the pandemic. Although subsequently recovering to the reference levels, the lengthened waiting times may be reflected in increased mortality and a need for more complex surgery in the near future.

## Supporting information

**S1 File. Anonymized data.** Anonymized data set supporting the findings of this study.
(XLSX)

**S2 File. Fig 1 point estimates.** Point estimates extracted from Fig 1.
(XLSX)

**S3 File. Fig 2 point estimates.** Point estimates extracted from Fig 2.
(XLSX)

**S4 File. Fig 3 point estimates.** Point estimates extracted from Fig 3.
(XLSX)

**S5 File. Fig 4 point estimates.** Point estimates extracted from Fig 4.
(XLSX)

## Author Contributions

**Conceptualization:** Mikko Uimonen, Ilari Kuitunen, Ville Ponkilainen, Ville M. Mattila.

**Data curation:** Mikko Uimonen, Ilari Kuitunen, Ville Ponkilainen.

**Formal analysis:** Mikko Uimonen.

**Investigation:** Mikko Uimonen, Ilari Kuitunen.

**Methodology:** Mikko Uimonen, Ville Ponkilainen.

**Project administration:** Ville M. Mattila.

**Supervision:** Ville M. Mattila.

**Validation:** Juha Paloneva, Antti P. Launonen, Ville Ponkilainen, Ville M. Mattila.

**Visualization:** Mikko Uimonen.

**Writing – original draft:** Mikko Uimonen.

**Writing – review & editing:** Ilari Kuitunen, Juha Paloneva, Antti P. Launonen, Ville Ponkilainen, Ville M. Mattila.

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
