## [Decision Letter · Decision Letter 0]

7 Jun 2021

PONE-D-21-12922

The Impact of the COVID-19 Pandemic on Waiting Times for Elective Surgery Patients: A Multicenter Study

PLOS ONE

Dear Dr. Uimonen,

Thank you for submitting your manuscript to PLOS ONE. After careful consideration, we feel that it has merit but does not fully meet PLOS ONE’s publication criteria as it currently stands. Therefore, we invite you to submit a revised version of the manuscript that addresses the points raised during the review process.

We look forward to receiving your revised manuscript.

Kind regards,

Corstiaan den Uil

Academic Editor

PLOS ONE

Journal Requirements:

3. In your ethics statement in the Methods section and in the online submission form, please provide additional information about the data used in your retrospective study. Specifically, please ensure that you have discussed whether all data were fully anonymized before you accessed them and/or whether the IRB or ethics committee waived the requirement for informed consent. If patients provided informed written consent to have data from their medical records used in research, please include this information.

4.We note that you have indicated that data from this study are available upon request. PLOS only allows data to be available upon request if there are legal or ethical restrictions on sharing data publicly. For information on unacceptable data access restrictions, please see http://journals.plos.org/plosone/s/data-availability#loc-unacceptable-data-access-restrictions.

Reviewers' comments:

Reviewer's Responses to Questions

**Comments to the Author**

1. Is the manuscript technically sound, and do the data support the conclusions?

Reviewer #1: Yes

2. Has the statistical analysis been performed appropriately and rigorously? 

Reviewer #1: Yes

3. Have the authors made all data underlying the findings in their manuscript fully available?

Reviewer #1: Yes

4. Is the manuscript presented in an intelligible fashion and written in standard English?

Reviewer #1: Yes

5. Review Comments to the Author

Reviewer #1: The findings reported in this extensive data base/registry analysis are logical and have in a similar way been presented elsewhere (and in other societies and health care systems); thus findings are not revolutionary or new, but still important to portrait the reaction of the system in Finland.

The manuscript would certainly benefit from inclusion of cardiovascular or at least cardiac surgeries and interventions which may have had a bigger impact onto morbidity/mortality than operations to the gastrointestinal or urogenital system.

Another peculiar observation is obviously a "summer brake phenomenon" clearly seen regardless of the pandemic in all the years analysed. This phenomenon should also be discussed as it deserves comments and could raise speculations.

Finally, it is a pity that the observations of the impact of Covid-19 in Finland could not be compared directly with mortality figures (that should by now be available at the National Office of Statistics).

Again, the information will probably not be really new but important for the country.

In their conclusion the authors should be more daring and suggest how to better react to a lockdown in a future pandemic, and how resources and capacity in the health care system should be allocated better,

6. PLOS authors have the option to publish the peer review history of their article (what does this mean?). If published, this will include your full peer review and any attached files.

Reviewer #1: **Yes: **Christoph A. Nienaber

---

## [Author Response · Author response to Decision Letter 0]

13 Jun 2021

Reviewer #1: 

The findings reported in this extensive data base/registry analysis are logical and have in a similar way been presented elsewhere (and in other societies and health care systems); thus findings are not revolutionary or new, but still important to portrait the reaction of the system in Finland.

The authors’ response: Thank you.

The manuscript would certainly benefit from inclusion of cardiovascular or at least cardiac surgeries and interventions which may have had a bigger impact onto morbidity/mortality than operations to the gastrointestinal or urogenital system.

The authors’ response: Thank you for this suggestion. We expanded the analysis to cover the surgeries performed due to cardiovascular diseases. Please see the corresponding revisions in Methods (line 70), Results (lines 92-96) and Discussion (lines 124-126) sections as well as in Figure 3.

Another peculiar observation is obviously a "summer brake phenomenon" clearly seen regardless of the pandemic in all the years analysed. This phenomenon should also be discussed as it deserves comments and could raise speculations.

The authors’ response: Thank you. Discussion on the summer brake phenomenon has been added (lines 114-122). 

Finally, it is a pity that the observations of the impact of Covid-19 in Finland could not be compared directly with mortality figures (that should by now be available at the National Office of Statistics).

Again, the information will probably not be really new but important for the country.

The authors’ response: Thank you for this suggestion. We added discussion on the mortality rates in Finland during the pandemic (lines 136-142) and Figure 4. However, since Statistics Finland database does not provide monthly mortality rates for the catchment areas of the study hospitals, we had to use monthly mortality data for the entire country. Nonetheless, we considered the data obtained from the study hospitals representative of the Finnish population. Therefore, the data of the study hospitals may be considered comparable to the national mortality rates. 

In their conclusion the authors should be more daring and suggest how to better react to a lockdown in a future pandemic, and how resources and capacity in the health care system should be allocated better. 

The authors’ response: Suggestions for preparation for future pandemics have been added to Discussion section (lines 132-152).

---

## [Editor Report · Decision Letter 1]

15 Jun 2021

The Impact of the COVID-19 Pandemic on Waiting Times for Elective Surgery Patients: A Multicenter Study

PONE-D-21-12922R1

Dear Dr. Uimonen,

We’re pleased to inform you that your manuscript has been judged scientifically suitable for publication and will be formally accepted for publication once it meets all outstanding technical requirements.

Kind regards,

Corstiaan den Uil

Academic Editor

PLOS ONE
---

## [Editor Report · Acceptance letter]

24 Jun 2021

PONE-D-21-12922R1 

The Impact of the COVID-19 Pandemic on Waiting Times for Elective Surgery Patients: A Multicenter Study 

Dear Dr. Uimonen:

I'm pleased to inform you that your manuscript has been deemed suitable for publication in PLOS ONE. Congratulations! Your manuscript is now with our production department. 

Kind regards, 

on behalf of

Dr. Corstiaan den Uil 

Academic Editor

PLOS ONE